# Agreement between Wedged Hepatic Venous Pressure and Portal Pressure in Hepatic Sinusoidal Obstruction Syndrome

**DOI:** 10.3390/jpm13010004

**Published:** 2022-12-20

**Authors:** Yang Cheng, Lihong Gu, Xiaochun Yin, Xixuan Wang, Jiangqiang Xiao, Yi Wang, Wei Zhang, Lei Wang, Xiaoping Zou, Ming Zhang, Yuzheng Zhuge, Feng Zhang

**Affiliations:** 1Department of Gastroenterology, Nanjing Drum Tower Hospital, The Affiliated Hospital of Nanjing University Medical School, Nanjing 210008, China; 2Department of Gastroenterology, Nanjing Drum Tower Hospital Clinical College of Jiangsu University, Nanjing 210008, China; 3Department of Gastroenterology, Medical School of Southeast University Nanjing Drum Tower Hospital, Nanjing 210008, China

**Keywords:** wedged hepatic vein pressure, portal pressure, hepatic venous pressure gradient, hepatic sinusoidal obstruction syndrome, portal hypertension

## Abstract

Background: Wedge hepatic vein pressure (WHVP) accurately estimates the portal pressure (PP) in chronic sinusoidal portal hypertension patients. Whether this applies to patients with acute portal hypertension due to hepatic sinusoidal obstruction syndrome (HSOS) is unclear. Our aim was to assess the agreement between WHVP and PP in patients with HSOS by comparing them to decompensated cirrhosis patients. Methods: From December 2013 to December 2021, patients with pyrrolidine alkaloid-induced HSOS (PA-HSOS) receiving hepatic venous pressure gradient (HVPG) measurement and transjugular intrahepatic portosystem shunt (TIPS) were retrospectively collected and matched with those of patients with virus- or alcohol-related cirrhosis as a cirrhosis group. Pearson’s correlation (R), intraclass correlation coefficient (ICC), scatter plots, and the Bland–Altman method were performed for agreement evaluation. Results: A total of 64 patients were analyzed (30 PA-HSOS and 34 cirrhosis groups). The correlation between WHVP and PP was moderate in the PA-HSOS group (R: 0.58, *p* = 0.001; ICC: 0.68, *p* = 0.002) but good in the cirrhosis group (R: 0.81, *p* < 0.001; ICC: 0.90, *p* < 0.001). The percentage of patients with inconsistent WHVP and PP in the two groups was 13 (43.3%) and 15 (26.5%) (*p* = 0.156), respectively, and an overestimation of PP was more common in the PA-HSOS group (33.3% vs. 2.9%, *p* = 0.004). HVPG and portal pressure gradient (PPG) consistency was poor in both groups (R: 0.51 vs. 0.26; ICC: 0.65 vs. 0.41; *p* < 0.05). Conclusions: WHVP in patients with PA-HSOS did not estimate PP as accurately as in patients with virus- or alcohol-related cirrhosis, which was mainly due to PP overestimation.

## 1. Introduction

Elevated portal vein pressure (PP) is the initiating factor for various complications associated with portal hypertension (PHT), including prehepatic, intrahepatic, and posthepatic portal hypertension [1], and PHT can be subdivided into acute, subacute, and chronic portal hypertension. Chronic portal hypertension is a common complication in patients with liver cirrhosis [2]. In contrast, acute or subacute portal hypertension is usually seen in hepatic vascular diseases, including hepatic sinusoidal obstruction syndrome (HSOS), Budd-Chiari syndrome, and acute portal vein thrombosis. The determination of the direct PP is invasive, risky, and technically demanding, and is not easily accepted by patients or medical personnel. Theoretically, the PP is higher than or equal to the hepatic sinusoidal pressure to maintain the portal venous flow to the liver. Typically, PP is indirectly represented by measuring the hepatic sinusoidal pressure, which can be obtained by measuring the wedged hepatic venous pressure (WHVP). Previous studies have shown that WHVP accurately estimates PP when liver disease affects the hepatic sinusoids, such as in chronic portal hypertension caused by cirrhosis [3,4,5]. Nevertheless, studies conducted to evaluate acute portal hypertension, such as that due to HSOS, have rarely been reported.

HSOS, previously named hepatic veno-occlusive disease, is a vascular liver disease characterized by swelling, necrosis, and detachment of endothelial cells in the hepatic sinusoids of the glandular follicle III region, leading to blockage and compression of the hepatic blood sinusoids [6], which is often caused by myeloablative pretreatment before hematopoietic stem cell transplantation (HSCT) in the West, whereas, in China, it is usually associated with oral intake of plants, such as ‘Tusanqi’ (Gynura segetum), that contain pyrrolizidine alkaloid (PA) [7]. Acute sinusoidal portal hypertension is its prominent clinical feature. Previous studies have demonstrated that an HVPG > 10 mmHg has a sensitivity of 52% and a specificity of 91% for the diagnosis of HSCT-HSOS [8]. The hepatic venous pressure gradient (HVPG) is the gold standard for the diagnosis of portal hypertension and is a valid indicator of indirect portal pressure levels, which are important for predicting complications and prognosis in patients with cirrhosis [9,10].

HVPG is calculated by subtracting the free hepatic venous pressure (FHVP) (a measure of systemic pressure) from WHVP (a measure of hepatic sinusoidal pressure) [9]. WHVP and PP are known to be in good agreement in viral- and alcohol-related cirrhosis [3,4,5]. However, the agreement between WHVP and PP has never been revealed in patients with HSOS. Therefore, the present study aimed to evaluate the accuracy of WHVP in estimating PP in patients with PA-HSOS.

## 2. Materials and Methods

### 2.1. Patients

This was a retrospective, single-center study that collected data from patients with PA-HSOS or alcohol- and virus-related cirrhosis undergoing a transjugular intrahepatic portosystemic shunt (TIPS) with concomitant HVPG measurements at the Department of Gastroenterology, Nanjing Drum Tower Hospital, from December 2013 to December 2021 (the cirrhotic patients served as the cirrhosis group). The time interval between the two procedures was within two days. The indications for TIPS in PA-HSOS patients are severe abdominal distension or an initial serum total bilirubin > 5 mg/dL, or a portal blood velocity < 10 cm/s. In patients with liver cirrhosis, the indications are refractory esophageal gastric variceal bleeding.

The patients’ intraoperative WHVP, free hepatic venous pressure (FHVP), inferior vena cava pressure (IVCP), PP, and other relevant data before shunting were recorded. HVPG and portal pressure gradient (PPG) were calculated. The inclusion criteria were as follows: (1) 18–75 years old; (2) diagnosed as PA-HSOS based on the Nanjing criteria for PA-HSOS [11] onset within 3 months or diagnosis as cirrhosis caused by viral or alcoholic hepatitis based on clinical, alcohol history, hepatitis virus serology, imaging features, or histological liver biopsy; (3) measurement of HVPG and PP during TIPS; and (4) provision of informed consent. The exclusion criteria were as follows: (1) prior liver transplantation; (2) severe portal vein thrombosis (>50% portal vein occlusion) and portal vein cavernous lesions; (3) the presence of hepatic vein-to-vein communications; and (4) the presence of malignant tumors or other severe diseases. The study complied with the Declaration of Helsinki and was approved by the Ethics Committee of Nanjing Drum Tower Hospital.

### 2.2. Intervention Procedure

The HVPG measurement and TIPS procedure were performed under local anesthesia, as previously described [12,13]. Briefly, the external zero reference point was set at the midaxillary line of the patient. The RUPS-100 (COOK, Bloomington, IN, USA) was placed in the inferior vena cava through the right internal jugular vein. A 5.5-7-F balloon-tipped catheter (Edwards Lifesciences, Irvine, CA, USA) was guided into the middle or the right hepatic vein. WHVP, FHVP, and IVCP were obtained, and the measurements were repeated three times. IVCP was measured near the entrance of the hepatic vein-inferior vena cava by withdrawing the floating catheter. After the pressure measurement was completed, a contrast agent was injected to confirm whether the obstruction was complete and the presence of hepatic vein-to-vein communications. After the HVPG measurement was completed, portal vein puncture was performed. A pigtail catheter was delivered into the main portal vein, and the PP was measured. HVPG is equal to WHVP minus FHVP, and portal pressure gradient (PPG) is equal to PP minus IVCP [9].

### 2.3. Definitions

Agreement between WHVP and PP occurs when the pressures are equal or differ by ≤10% of the PP value. Disagreement occurs when WHVP and PP differ by >10% of the PP value [14]. When WHVP is more than 10% lower than PP, it is defined as PP underestimated by WHVP. Conversely, when the latter is more than 10% higher than the former, the PP is overvalued. A difference of ≥5 mmHg between these two pressures was considered a major discrepancy [15].

### 2.4. Statistical Analysis

All statistical analyses were conducted by IBM SPSS Statistics version 22.0 (IBM, Armonk, NY, USA). The chi-square test or Fisher’s exact test was used to compare categorical variables, which were presented as numbers (N) with percentages (%). Continuous variables are presented as the median and interquartile range (25th to 75th percentile) and were compared by the Mann–Whitney U test. The correlation between WHVP and PP was assessed using scatter plots and the Bland–Altman plot. Pearson’s correlation coefficient (R), intraclass correlation coefficient (ICC), and confidence interval (CI) were calculated. A two-sided *p* value ≤ 0.05 was considered statistically significant.

## 3. Results

### 3.1. Patient Characteristics

The study included two cohorts of 64 patients, 30 with PA-HSOS (PA-HSOS group), 20 with virus-related cirrhosis, and 14 with alcoholic cirrhosis (cirrhosis group), according to the inclusion and exclusion criteria. Technical success was achieved in all patients with no complications related to the procedure. The baseline characteristics of the patients are summarized in Table 1. Of the patients with PA-HSOS, 21 (70%) were men and 9 (30%) were women. Ascites developed in all the patients, and 28 (93.3%) patients had moderate to severe ascites. The manometry results of all the patients in both groups indicated obvious portal hypertension (HVPG: 21.8 mmHg vs. 20.0 mmHg, *p* = 0.042). There was no significant difference in WHVP and PP between the two groups (29.0 mmHg vs. 29.3 mmHg; 28.0 mmHg vs. 29.5 mmHg; *p* > 0.05).

### 3.2. Correlation between WHVP and PP

Table 2 and Figure 1A,B show the correlation between WHVP and PP in the two groups of patients. The correlation between WHVP and PP in the patients with decompensated cirrhosis was remarkable (R: 0.81; ICC: 0.90; *p* < 0.001), as well as in the stratified analysis of the patients with viral cirrhosis (R: 0.85; ICC: 0.92; *p* < 0.001) or alcohol-related cirrhosis (R: 0.82; ICC: 0.90; *p* < 0.001). However, the correlation was moderate in patients with PA-HSOS (R: 0.58; ICC: 0.68; *p* = 0.002). The Bland and Altman graph showed that the evaluation of the agreement between WHVP and PP was good, and its 95% agreement interval compared with the PA-HSOS group range was even smaller, which confirms the numerical 1:1 correlation of WHVP with PP in cirrhosis (Figure 1C,D). As shown in Table 3, the agreement between WHVP and PP (differences < 10% of PP value) occurred in 17 (56.7%) and 25 (73.5%) patients in the PA-HSOS group and the cirrhosis group, respectively (*p* = 0.156).

Univariate analysis incorporating sex, age, etiology, hypertension, diabetes, ascites, portal velocity, total bilirubin (TB), albumin (Alb), and international normalized ratio (INR) was used to identify factors associated with the divergence between WHVP and PP, which was only associated with the PA-HSOS etiology and hypertension [OR: 0.11 (95% CI 0.01–1.00); *p* = 0.049; vs. OR: 0.19 (95% CI 0.04–0.83); *p* = 0.027]. The major discrepancies between WHVP and PP in the two groups were 4 (13.3%) and 5 (14.7%), respectively (*p* = 1.00). Among the disagreements between WHVP and PP in the two groups, the underestimation of PP was 3 (10.0%) and 8 (23.5%) patients, respectively (*p* = 0.271), while the overestimation of PP was 10 (33.3%) and 1 (2.9%) patients, respectively (*p* = 0.004), indicating that the WHVP in PA-HSOS was more inclined to overestimate PP.

### 3.3. Correlation between FHVP and IVCP

Table 4 and Figure 2A,B show the correlation between FHVP and IVCP in the two groups of patients. The correlation between FHVP and IVCP in the cirrhosis group was excellent (R: 0.89; ICC: 0.94; *p* < 0.001). Moreover, the correlations were also good in the patients with viral cirrhosis (R: 0.90; ICC: 0.92; *p* < 0.001) and alcohol cirrhosis (R: 0.84; ICC: 0.91; *p* < 0.001). Nevertheless, the correlation was moderate in the PA-HSOS group (R: 0.56; ICC: 0.71; *p* = 0.001). Moreover, as shown in Figure 2C,D, the individual variability of the cirrhosis group was lower than that of the PA-HSOS group.

### 3.4. Correlation between HVPG and PPG

The correlation between HVPG and PPG was poor in the cirrhosis group (R: 0.26; ICC: 0.41; *p* = 0.074), which was consistent with the results in patients with hepatitis viral cirrhosis or alcoholic cirrhosis, while, in the PA-HSOS group, the correlation was moderate (R: 0.51; ICC: 0.65; *p* = 0.003)), as Table 5 shows.

## 4. Discussion

Our findings, for the first time, suggest that the agreement between WHVP and PP in PA-HSOS patients is only moderate and is not as good as in hepatitis virus cirrhosis or alcoholic cirrhosis, while the higher concordance between the latter two was consistent with national and international reports [3,4,5]. Disagreement between WHVP and PP was more frequently seen when PP was overestimated in patients with PA-HSOS; whereas, in hepatitis viral cirrhosis or alcoholic cirrhosis, the PP tends to be slightly underestimated.

According to previous reports, the etiology of HSOS includes pretreatment with a large number of chemicals before HSCT; adjuvant chemotherapy containing oxaliplatin; immunosuppressive use after liver transplantation; autosomal recessive veno-occlusive disease with immunodeficiency; ingestion of plants containing PAs, etc. [16]. The majority of HSOS patients in Western countries occurred after HSCT, which is associated with factors such as high-dose chemotherapeutic drug pretreatment. Whereas there are few reports of HSCT–HSOS in China, and the percentage of HSOS caused by the use of “Tusanqi” is 50.0% to 88.6% [11]. It is well known that the higher the PP, the more severe the associated complications, such as abdominal distention, liver pain, ascites, and other clinical symptoms. Thus, treatment to reduce portal vein pressure may provide multiple benefits to patients. 

Anticoagulation-TIPS step therapy is the most common treatment modality for acute or subacute PA-HSOS [8,17,18,19]. In addition, previous studies have shown that TIPS reverses progression in patients who have failed anticoagulation for PA-HSOS, with significant reductions in PP after TIPS treatment and survival rates of 80–90% [17,20]. Early assessment of the degree of portal hypertension in patients can better guide individualized treatment. For high-risk patients, timely TIPS treatment may lead to a better prognosis.

Direct measurement of PP is not easily accepted by patients and does not accurately predict the risk of developing various PHT complications [21,22,23]. Thus, researchers have searched for a simple, easy-to-use, and patient-friendly method to accurately reflect PPG. In 1953, Krook first proposed WHVP as an indirect measurement of PP and correlated it to the magnitude of PP values, and measuring WHVP is very safe [24]. WHVP accurately estimates PP in sinusoidal portal hypertension. However, previous studies have been mainly conducted in patients with cirrhosis. Unlike chronic portal hypertension resulting from liver cirrhosis, in patients with acute portal hypertension, such as HSOS, the main clinical manifestations are abdominal distention, jaundice, and refractory ascites, but collateral circulation and splenomegaly are not seen [25]. The pathological change in HSOS is endothelial injury in the sinusoids, predominantly in the centrilobular areas [11,26], which have similar pathological features regardless of the cause.

Our previous study investigated 116 patients with PA-HSOS who underwent TIPS and found that the mean PPG was as high as 21.48 mmHg [27]. Patients with PA-HSOS needing to receive TIPS all had refractory ascites, which could falsely elevate WHVP [9]. On the other hand, it often means that there are postsinusoidal components contributing to portal hypertension if WHVP overestimates PP, such as in Budd-Chiari Syndrome (BCS). Although PA-HSOS is always considered sinusoidal portal hypertension according to the pathological change of endothelial injury, it might be accompanied by postsinusoidal injury in the central vein because PAs could damage many kinds of vascular endothelium. In general, WHVP measurements appear to slightly underestimate PP in viral- or alcohol-related cirrhosis, but the difference is small (1.3 mmHg), possibly due to the umbilical vein opening, portal anastomotic branch, gastric shunt, or the presence of tiny hepatic venous-to-venous communication branches that were not detected or reversed liver blood flow, etc. [15,18,28].

In this study, the agreement between WHVP and PP and FHVP and IVCP was very good, but the difference value between the two (HVPG and PPG, respectively) was not significantly correlated. In other words, HVPG cannot accurately reflect the degree of portal hypertension and needs to be evaluated together with other examinations. In our study, PP was directly measured during TIPS, and there are still multiple factors that can affect the accuracy of the measurement. In the study, HVPG measurement and the TIPS procedure were performed under local anesthesia. Usually, patients have no obvious discomfort in the process of measuring HVPG. However, some patients cannot cooperate due to the pain associated with the process of puncturing the portal vein, and clinicians often give sedative and analgesic drugs such as morphine or pethidine. A previous study demonstrated that the use of propofol for deep sedation, however, has an effect on the patients’ WHVP, FHVP, IVCP, PP, and PPG, but the effect on PPG is greater [29]. Moreover, most of the patients who came to the hospital with gastrointestinal bleeding had taken preoperative PP-lowering drugs, such as growth inhibitors and their analogs, posterior pituitary leptin, terlipressin, etc. The effect of these drugs on the abovementioned manometry is unclear and needs to be further explored in the future.

The main limitation of this study is that it is a single-center retrospective study and has a small sample size. A higher rate (23.8%) of underestimation of PP in HBV and alcohol-related cirrhosis was observed in this study, as compared to a previous study that had a 7.5% underestimation in HCV and alcohol-related cirrhosis [14]. The potential reason might be the selection bias due to the small sample size. In this study, the degree of ascites in the two groups of patients was unbalanced. Whether this will affect the consistency of WHVP and PP needs further research based on balanced baseline data of mild ascites in the future.

In conclusion, the agreement between WHVP and PP in patients with PA-HSOS is lower than in patients with viral- or alcohol-related cirrhosis, mainly due to PP overestimation.

## Figures and Tables

**Figure 1 jpm-13-00004-f001:**
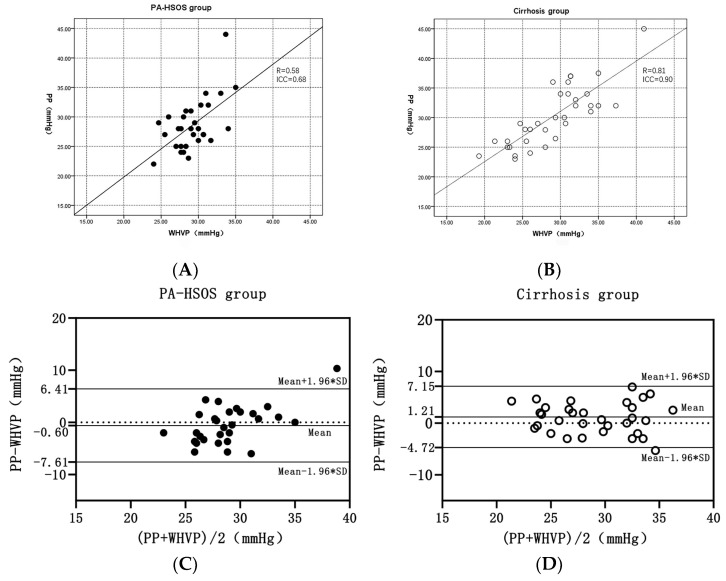
Correlation between WHVP and PP according to a scatter plot and the Altman and Bland plot. (**A**). In the patients with PA-HSOS (*n* = 30), the strength of the linear correlation between WHVP and PP was moderate (R: 0.58; ICC: 0.68). (**B**). In the patients with cirrhosis (*n* = 34), the strength of the linear correlation between WHVP and PP was excellent (R: 0.81; ICC: 0.90). (**C**). In the PA-HSOS group (*n* = 30), one point was not within 95% of the mean difference between WHVP and PP, which ranged from −7.61 to 6.41 mmHg. (**D**). In the cirrhosis group (*n* = 34), two points were not within 95% of the mean difference between WHVP and PP, which ranged from −4.72 to 7.15 mmHg.

**Figure 2 jpm-13-00004-f002:**
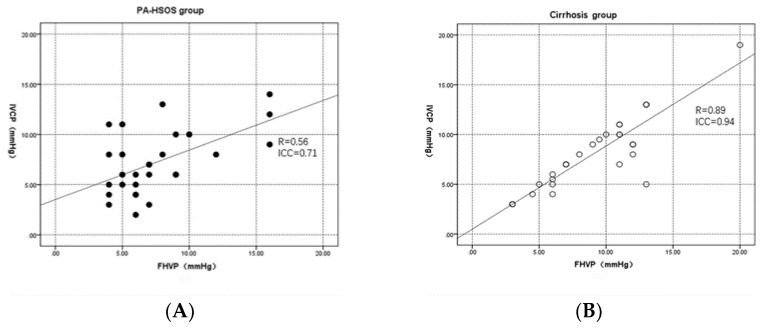
Correlation between FHVP and IVCP according to the scatter plot and the Altman and Bland plot. (**A**). In the patients with PA-HSOS (*n* = 30), the strength of the linear correlation between FHVP and IVCP was moderate (R: 0.56; ICC: 0.71). (**B**). In the patients with cirrhosis (*n* = 34), the strength of the linear correlation between WHVP and PP was excellent (R: 0.89; ICC: 0.94). (**C**). In the PA-HSOS group (*n* = 30), three points were not within 95% of the mean difference between WHVP and PP, which ranged from −6.43 to 5.76 mmHg. (**D**). In the cirrhosis group (*n* = 34), nine points were not within 95% of the mean difference between WHVP and PP, which ranged from −3.93 to 1.87 mmHg.

**Table 1 jpm-13-00004-t001:** Baseline characteristics of the patients.

Patient Characteristics	PA-HSOS (*n* = 30)	Decompensated Cirrhosis (*n* = 34)	*p*-Value
Age (years)	65 (59–68)	58 (51–66)	0.005
Male	21 (70%)	25 (73%)	0.294
Arterial hypertension	13 (43%)	8 (24%)	0.092
Type 2 diabetes	7 (23%)	7 (21%)	0.791
Severity of liver disease			
Ascites, no/light/medium/heavy	0/2/24/4	6/11/13/4	0.002
Portal velocity(cm/s)	13.7 (11.1–20.1)	31.0 (21.3–41.3)	<0.001
TB(µmol/L)	44.3 (30.7–77.3)	18.3 (14.3–29.2)	<0.001
Alb (g/L)	32.4 (30.7–34.9)	32.9 (31.1–36.4)	0.633
Scr (µmol/L)	72.0 (60.8–103.3)	69.0 (53.8–87.8)	0.329
INR	1.3 (1.2–1.5)	1.3 (1.2–1.4)	0.477
Hepatic hemodynamics			
WHVP (mmHg)	29.0 (27.7–30.8)	29.3 (25.2–32.0)	0.888
PP (mmHg)	28.0 (25.7–31.0)	29.5 (26.0–34.0)	0.195
PP-WHVP (mmHg)	−0.8 (−3.7–+1.8)	1.3 (−1.2–+4.0)	0.017
FHVP(mmHg)	7.0 (5.0–9.0)	8.0 (6.0–11.5)	0.120
IVCP(mmHg)	7.0 (5.0–10.0)	8.0 (5.0–10.2)	0.354
HVPG(mmHg)	21.8 (18.9–24.4)	20.0 (17.8–22.5)	0.042
PPG(mmHg)	22.0 (18.0–24.2)	22.5 (20.0–24.4)	0.370

Data are expressed as median (IQR) or frequencies (%) as appropriate. Abbreviations. WHVP: wedge hepatic vein pressure; PP: portal pressure; FHVP: free hepatic vein pressure; IVCP: inferior vena cava pressure; HVPG: hepatic venous pressure gradient; PPG: portal pressure gradient; TB: total bilirubin; Alb: albumin; Scr: serum creatinine; INR: international normalized ratio; Ascites: (1) light: patients generally have abdominal distension, the ascites can only be detectable by an ultrasound examination, and the depth is <3 cm. (2) Medium: the patient often has moderate and symmetrical abdominal distension, and the depth is 3–10 cm. (3) Heavy: the patient has a significant bloating. The ascites detected by ultrasound occupies the entire abdominal cavity, and the depth is >10 cm.

**Table 2 jpm-13-00004-t002:** Correlation between WHVP and PP.

	R	95% CI	*p*	ICC	95% CI	*p*
PA-HSOS group (*n* = 30)	0.58	0.25–0.77	0.001	0.68	0.32–0.85	0.002
Cirrhosis group (*n* = 34)	0.81	0.68–0.90	<0.001	0.90	0.79–0.95	<0.001
Viral-related cirrhosis (*n* = 20)	0.85	0.73–0.92	<0.001	0.92	0.79–0.97	<0.001
Alcohol-related cirrhosis (*n* = 14)	0.82	0.38–0.97	<0.001	0.90	0.69–0.97	<0.001

Abbreviations. WHVP: wedge hepatic vein pressure; PP: portal pressure; R: Pearson’s correlation; ICC: Interclass correlation coefficient; CI: confidence interval; PA-HSOS: Pyrrole alkaloid-associated sinusoidal occlusion syndrome.

**Table 3 jpm-13-00004-t003:** Performance of WHVP in evaluating PP.

Patient Characteristics	PA-HSOS Group (*n* = 30) *n* (%)	Cirrhosis Group (*n* = 34) *n* (%)	*p*-Value
Agreement between WHVP and PP	17 (56.7%)	25 (73.5%)	0.156
Disagreement between WHVP and PP	13 (43.3%)	9 (26.5%)	
Underestimation of PP	3 (10.0%)	8 (23.5%)	0.271
Overestimation of PP	10 (33.3%)	1 (2.9%)	0.004
Major discrepancies between WHVP and PP	4 (13.3%)	5 (14.7%)	1.000

Abbreviations. WHVP: wedge hepatic vein pressure; PP: portal pressure; PA-HSOS: Pyrrole alkaloid-associated sinusoidal occlusion syndrome.

**Table 4 jpm-13-00004-t004:** Correlation between FHVP and IVCP.

	R	95% CI	*p*	ICC	95% CI	*p*
PA-HSOS group (*n* = 30)	0.56	0.22–0.80	0.001	0.71	0.40–0.86	0.001
Cirrhosis group (*n* = 34)	0.89	0.68–0.98	<0.001	0.94	0.87–0.97	<0.001
Viral-related cirrhosis (*n* = 20)	0.90	0.82–0.97	<0.001	0.92	0.69–0.97	<0.001
Alcohol-related cirrhosis (*n* = 14)	0.84	0.12–1.00	0.01	0.91	0.67–0.97	<0.001

Abbreviations. FHVP: Free hepatic vein pressure; IVCP: Inferior vena cava pressure; R: Pearson’s correlation; ICC: Interclass correlation coefficient; CI: Confidence interval; PA-HSOS: Pyrrole alkaloid-associated sinusoidal occlusion syndrome.

**Table 5 jpm-13-00004-t005:** Correlation between HVPG and PPG.

	R	95% CI	*p*	ICC	95% CI	*p*
PA-HSOS group (*n* = 30)	0.51	0.17–0.78	0.004	0.65	0.27–0.84	0.003
Cirrhosis group (*n* = 34)	0.26	−0.25–0.55	0.156	0.41	−0.21–0.72	0.074
Viral-related cirrhosis (*n* = 20)	0.38	0.00–0.66	0.125	0.55	−0.21–0.83	0.057
Alcohol-related cirrhosis (*n* = 14)	0.38	−0.32–0.77	0.209	0.38	−0.32–0.77	0.096

Abbreviations. HVPG: Hepatic venous pressure gradient; PPG: Portal pressure gradient; R: Pear-son’s correlation; ICC: Interclass correlation coefficient; CI: Confidence interval; PA-HSOS: Pyrrole alkaloid-associated sinusoidal occlusion syndrome.

## Data Availability

Corresponding authors can be contacted for relevant data upon reasonable request.

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
