# Peer review of "Agreement between Wedged Hepatic Venous Pressure and Portal Pressure in Hepatic Sinusoidal Obstruction Syndrome"

_jpm, 2022, doi:10.3390/jpm13010004_

Round 1

Reviewer 1 Report

Accurate assessment of portal pressure in patients with chronic liver disease and portal hypertension is important in terms of correct diagnosis and defining indications for treatment. Liver vein catheterization with measuring the hepatic venous pressure gradient as a difference between the wedged hepatic venous pressure and the free hepatic venous pressure is the gold standard and an appropriate method in most patients with portal hypertension. However, the method depends on the site of hepatic obstruction as being pre-, intra, or post hepatic portal hypertension. In some patients with sinusoidal obstruction, previously known as veno-occlusive disease the method may be less accurate because of discrepancies between the pressure in the portal vein and the wedged hepatic venous pressure as an expression n of the portal venous pressure. In the present study, the authors have compared measurements of portal pressure – wedged hepatic venous pressure differences in a group of 30 patients with hepatic sinusoidal obstruction syndrome with a group of 34 patients with cirrhosis. All patients underwent a TIPS-procedure that allowed measurements of the direct portal venous pressure. Main results were that this mean difference differed in HSOS-patients (-0.8 mmHg) compared to 1.3 mmHg) in cirrhotic patients. This reviewer agrees that it is important to be aware that the HVPG may be slightly overestimated in HSOS-patients by approximately 1 mmHg. This is an interesting and relevant study but this reviewer has some comments for the authors consideration.

1.       The authors should provide a clear definition of HSOS in accordance with current guidelines.

2.       The problem with pyrrolizidine alcaloide intoxication is less prevalent in Western countries but the same may hold true regarding hematological disease (mastocytosis, granulomas, scistosomiasis and others. Please discuss.

3.       Can the authors discuss which type of histological features that determine differences in the portal venous pressure and the wedged hepatic venous pressure?

4.       Although statistical significant (-0.3 vs. 1.3 mmHg) this difference may not be clinically different. Please discuss.

5.       How do the authors explain differences in the correlation coefficients from Table 3 (0.58 vs 0.81)?

6.       What are the coefficients of variation between your measurements of PP and WHVP and are there differences between the two groups?

7.       If you have a patient with HSOS and suspect an erroneously high WHVP, what is then your alternative to the current measurement?

8.       Please discuss the clinical consequences of your study.

Author Response

The point-by-piont responses to the reviewers’ comments:

First of all, we thank the reviewers for their positive and constructive comments and suggestions.

Reviewer #1

Major points:

Accurate assessment of portal pressure in patients with chronic liver disease and portal hypertension is important in terms of correct diagnosis and defining indications for treatment. Liver vein catheterization with measuring the hepatic venous pressure gradient as a difference between the wedged hepatic venous pressure and the free hepatic venous pressure is the gold standard and an appropriate method in most patients with portal hypertension. However, the method depends on the site of hepatic obstruction as being pre-, intra, or post hepatic portal hypertension. In some patients with sinusoidal obstruction, previously known as veno-occlusive disease the method may be less accurate because of discrepancies between the pressure in the portal vein and the wedged hepatic venous pressure as an expression n of the portal venous pressure. In the present study, the authors have compared measurements of portal pressure – wedged hepatic venous pressure differences in a group of 30 patients with hepatic sinusoidal obstruction syndrome with a group of 34 patients with cirrhosis. All patients underwent a TIPS-procedure that allowed measurements of the direct portal venous pressure. Main results were that this mean difference differed in HSOS-patients (-0.8 mmHg) compared to 1.3 mmHg) in cirrhotic patients. This reviewer agrees that it is important to be aware that the HVPG may be slightly overestimated in HSOS-patients by approximately 1 mmHg. This is an interesting and relevant study but this reviewer has some comments for the authors consideration.

  • The authors should provide a clear definition of HSOS in accordance with current guidelines.

Response: We highly appreciate your good suggestion. According to your consideration, the precise definition of HSOS has been added to the introduction of the revision, citing the expert consensus published in J Gastroenterol Hepatol on the " Nanjing criteria" for the diagnosis of PA-HSOS.

1.2 The problem with pyrrolizidine alcaloide intoxication is less prevalent in Western countries but the same may hold true regarding hematological disease (mastocytosis, granulomas, schistosomiasis and others. Please discuss.

Response: Thank you very much for your comment. The etiologies of HSOS include cytoreductive therapy prior to hematopoietic stem cell transplantation (HSCT); oxaliplatin containing adjunctive chemotherapy; ingestion of plants containing pyrrolizidine alkaloids (PAS); use of tacrolimus in liver transplantation; and autosomal recessive occlusive veno-occlusive disease combined with immunodeficiency etc. In western countries, HSOS usually occurs in patients who have received cytoreductive therapy prior to HSCT or oxaliplatin-containing chemotherapy for colorectal carcinoma. In China, the primary cause of HSOS is the ingestion of PAs-containing herbals or dietary supplements. However, mastocytosis, granulomas, and schistosomiasis have rarely been reported to be associated with the occurrence of HSOS.

1.3 Can the authors discuss which type of histological features that determine differences in the portal venous pressure and the wedged hepatic venous pressure?

Response: Thank you for your suggestion. Theoretically, PP is higher or equal to hepatic sinusoidal pressure to maintain portal venous to hepatic blood flow, and if PP is lower than hepatic sinusoidal pressure, a reverse hepatic blood flow. Therefore, it is usually possible to indirectly represent PP by measuring hepatic sinusoidal pressure.

WHVP accurately estimates PP when liver disease destroys the sinusoids as it happens in cirrhosis due to alcohol or viral hepatitis. The reason for WHVP being higher than PP is not clear. Some studies have found that patients with WHVP higher than PP tend to have reverse hepatic flow, open accessory umbilical veins, and portal anastomotic branches. It has also been found that these patients tend to have gastric-renal shunts.

1.4 Although statistical significant (-0.3 vs. 1.3 mmHg) this difference may not be clinically different. Please discuss.

Response: Thank you very much for your comment. The median difference between WHVP and PP was higher in the cirrhosis group than in the PA-HSOS group [1.3(-1.2 – +4.0) vs 0.8 (-3.7 –+1.8) mmHg; p=0.017 ], suggesting that in patients with cirrhosis, WHVP tends to underestimate PP.

1.5 How do the authors explain differences in the correlation coefficients from Table 3 (0.58 vs 0.81)?

Response: Thank you very much for your comment. The correlation between WHVP and PP in the patients with decompensated cirrhosis was remarkable(R: 0.81), whereas, it was moderate in patients with PA-HSOS((R: 0.58). The correlation between WHVP and PP was significant in patients with decompensated cirrhosis (R: 0.81) and moderate in PA-HSOS patients (R: 0.58). In other words, WHVP can accurately assess PP in patients with cirrhosis, but not so accurately in PA-HSOS, as described in the results and discussion section of the paper.

1.6 What are the coefficients of variation between your measurements of PP and WHVP and are there differences between the two groups?

Response: The coefficients of variation of WHVP and PP in PA-HSOS and cirrhosis groups were 9.1% vs. 16.7% and 46.0% vs. 42.5%, respectively.

1.7 If you have a patient with HSOS and suspect an erroneously high WHVP, what is then your alternative to the current measurement?

Response: The main purpose of HVPG measurement is to assist diagnosis, evaluation of disease severity and prognosis. When the measured WHVP is considered to be incorrectly elevated, on the one hand, transjugular liver biopsy can be performed during HVPG measurement to facilitate pathological diagnosis. On the other hand, Other auxiliary tests such as AST, TB, FIB, and portal vein flow rate can be used to calculate the Drum Tower Severity score (DTSS) to assist in the assessment of disease severity and prognosis.

1.8 Please discuss the clinical consequences of your study.

Response: Thank you for your comment. In patients with sinusoidal portal hypertension, the use of HVPG relies on good agreement between direct measurements of PP and indirect measurements of WHVP. However, our study found this consensus to be less accurate in PA-HSOS, a non-cirrhotic portal hypertensive disease. Under normal conditions, HVPG does not exceed 5mmHg. Once the HVPG exceeds 10mmHg, it is clinically significant portal hypertension. If the HVPG exceeds 12mmHg, there is a risk of esophageal varices rupture and bleeding. However, our study suggests that the threshold for HVPG prediction outcomes may be different in the PA-HSOS population and warrants specific research. In addition, in HSOS, HVPG cannot accurately reflect the degree of portal hypertension and needs to be evaluated together with other examinations.

Reviewer 2 Report

Comments to the paper entitled “Agreement between wedged hepatic venous pressure and portal pressure in hepatic sinusoidal obstruction syndrome” to be published in Journal of Personalized Medicine.

In this elegant radiological study, the authors concluded that wedged hepatic venous pressure measurement in patients with pyrrolidine alkaloid-induced portal hypertension did not estimate portal pressure as accurately as in patients with virus- or alcohol-related cirrhosis, which was mainly due to PP overestimation.

In fact, the study is very limited in conclusions.

My questions:

What is the clinical importance of such results?

What are the clinical implications of the results?

What the therapeutic implications of the results?

Do the encountered results and conclusions have any implication in treatment?

These answers are important to be responded in discussion.

Author Response

The point-by-piont responses to the reviewers’ comments:

First of all, we thank the reviewers for their positive and constructive comments and suggestions.

Reviewer #2

Major comments:

In this elegant radiological study, the authors concluded that wedged hepatic venous pressure measurement in patients with pyrrolidine alkaloid-induced portal hypertension did not estimate portal pressure as accurately as in patients with virus- or alcohol-related cirrhosis, which was mainly due to PP overestimation.

In fact, the study is very limited in conclusions.

2.1 What is the clinical importance of such results?

Response: Thank you very much for your comment. Although HVPG is a very active area of research for portal hypertension, knowledge about prognostic predictors in patients with PA-HSOS remains poor. In fact, we know very little about the predictive role of HVPG, which has been demonstrated in viral hepatitis and alcoholic hepatitis cirrhosis. In addition, some foreign countries have reported that HVPG > 10 mm Hg has a sensitivity of 52% and a specificity of 91%. HVPG also has some prognostic value. However, the use of HVPG relies on the high agreement between direct measurement of PP and indirect measurement of WHVP in patients with sinusoidal portal hypertension. Our findings, for the first time, suggest that the agreement between WHVP and PP in PA-HSOS patients is only moderate and is not as good as in hepatitis virus cirrhosis or alcoholic cirrhosis. These results will open a new window for future studies aimed at evaluating the role of WHVP in assessing PP in patients with PA-HSOS, including diagnosis, guiding treatment modalities and prognostic survival value.

2.2 What are the clinical implications of the results?

Response: Thank you for your comment. In patients with sinusoidal portal hypertension, the use of HVPG relies on good agreement between direct measurements of PP and indirect measurements of WHVP. However, our study found this consensus to be less accurate in PA-HSOS, a non-cirrhotic portal hypertensive disease. Under normal conditions, HVPG does not exceed 5mmHg. Once the HVPG exceeds 10mmHg, it is clinically significant portal hypertension. If the HVPG exceeds 12mmHg, there is a risk of esophageal varices rupture and bleeding. However, our study suggests that the threshold for HVPG prediction outcomes may be different in the PA-HSOS population and warrants specific research. In addition, in HSOS, HVPG cannot accurately reflect the degree of portal hypertension and needs to be evaluated together with other examinations.

2.3 What the therapeutic implications of the results? Do the encountered results and conclusions have any implication in treatment?

Response: Thank you very much for your valuable comment. It is well known that the primary purpose of measuring HVPG is to assist in the diagnosis and assessment of disease severity and prognosis. We found that the agreement between WHVP and PP in patients with PA-HSOS is lower than in patients with viral or alcohol-related cirrhosis, mainly due to PP overestimation. WHVP and HVPG thresholds for the clinical prognosis of patients with PA-HSOS may need to be redefined, and the choice of treatment modality needs to be guided by other adjuvant indicators in combination with others.

Reviewer 3 Report

In the presented manuscript, Yang Cheng. et al. have analyzed the correlation between wedged hepatic venous pressure and portal pressure in hepatic sinusoidal obstruction syndrome.

This article presents new insights into complications of cholestatic disorders. 

The introduction is well-written. The discussion section needs to be revised and needs to be improved. Please rearrange this section for better understanding and make it easier to follow the researcher's thoughts. The discussion needs to be written more consistently, which makes it difficult to read. It should also indicate which information in the literature is consistent and contrary to the obtained results. Also, the discussion section lacks a part with the limitation of this study. In the discussion section, literature concerning the analyzed factors is nicely cited.

Author Response

We highly appreciate your good suggestion. According to your consideration, we have added this information to the discussion section accordingly.

Round 2

Reviewer 3 Report

Thank you for added information. Now the article looks good.